# Animatable 3D Gaussian: Fast and High-Quality Reconstruction of Multiple Human Avatars

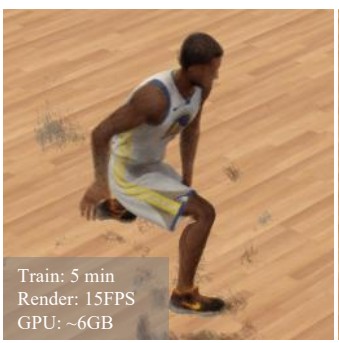 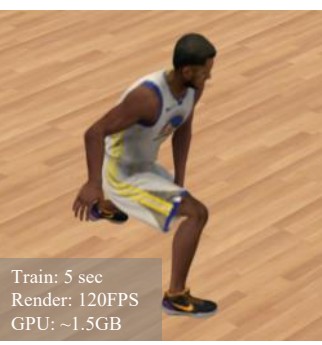 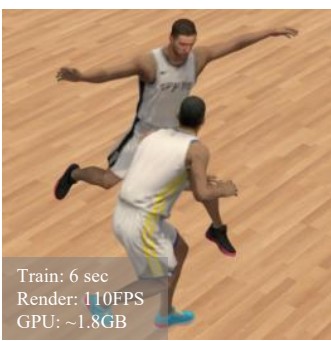 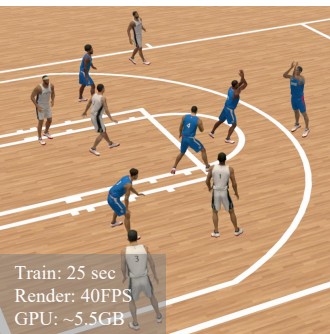

| InstantAvatar | Ours | Ours: Double-Human | Ours: Multi-Human |

**Figure 1: Novel View Synthesis from Multi-View Video. We present novel view synthesis results of our proposed Animatable 3D Gaussian on single-human, double-human, and multi-human scenes. Our method can produce higher quality synthesis results than InstantAvatar [26] with only a few seconds of training time, and render novel view images at real-time speed. Moreover, our method requires a very small amount of GPU memory. We implement all experiments only on a single RTX 3090.**

## ABSTRACT

Neural radiance fields are capable of reconstructing high-quality drivable human avatars but are expensive to train and render and not suitable for multi-human scenes with complex shadows. To reduce consumption, we propose Animatable 3D Gaussian, which learns human avatars from input images and poses. We extend 3D Gaussians to dynamic human scenes by modeling a set of skinned 3D Gaussians and a corresponding skeleton in canonical space and deforming 3D Gaussians to posed space according to the input poses. We introduce a multi-head hash encoder for pose-dependent shape and appearance and a time-dependent ambient occlusion module to achieve high-quality reconstructions in scenes containing complex motions and dynamic shadows. On both novel view synthesis and novel pose synthesis tasks, our method achieves higher reconstruction quality than InstantAvatar with less training time (1/60), less GPU memory (1/4), and faster rendering speed (7×). Our method can be easily extended to multi-human scenes and achieve comparable novel view synthesis results on a scene with ten people in only 25 seconds of training. We will release the code and dataset.

## CCS CONCEPTS

• **Computing methodologies → Shape representations**.

## KEYWORDS

Human Reconstruction, 3D Gaussian Splatting, Neural Radiance field

## 1 INTRODUCTION

Real-time rendering and fast reconstruction of a high-quality digital human has a variety of applications in various fields, such as virtual reality, gaming, sports broadcasting, and telepresence. Existing methods often take considerable time for training, and none of them can achieve high-quality reconstruction and real-time rendering in a very short training time.

Recent methods [5, 26, 37] implicitly reconstruct high-quality human avatars using neural radiance fields [35]. However, implicit neural radiance fields inevitably lead to artifacts in the synthesized novel views as they require modeling the entire space including empty space. Some of these methods [5, 37] require high memory and time consumption due to multilayer perceptron and complex point sampling for each ray. When it comes to datasets with dynamic illumination and shadow, the reconstruction quality of these methods [5, 26, 37] is significantly degraded. This decline is attributed to the inherent incapacity of these methods to accommodate the dynamic alterations in illumination and shadow.

In this paper, we aim to acquire high-fidelity human avatars from a monocular or sparse-view video sequence in seconds and to render high-quality novel views and poses at interactive rates (170 FPS for $540 \times 540$ resolution). To this end, we introduce a novel neural representation using 3D Gaussians (3D-GS [28]) for dynamic humans, named Animatable 3D Gaussian, overcoming the problem of artifacts in implicit neural radiance fields. In order to deform 3D Gaussians to the posed space, we model a set of skinned 3D Gaussians and a corresponding skeleton in canonical space. We use the multi-head hash encoder to encode the shape and appearance of 3D

Gaussians to accelerate convergence speed, avoid overfitting, and acquire pose-dependent deformation. For low memory and time consumption of rendering, we use the 3D Gaussian rasterizer [28] to rasterize the deformed 3D Gaussians instead of volume rendering. To handle the dynamic illumination and shadow, we suggest modeling time-dependent ambient occlusion for each timestamp.

Since the public dataset [1, 37] contains few pose and shadow changes, we create a new dataset named GalaBasketball in order to show the performance of our method under complex motion and dynamic shadows. We evaluate our method on both public and our created dataset and compare it with state-of-the-art methods [5, 26]. Our method is able to reconstruct better-quality human avatars in a shorter time. Moreover, our method can be extended to multi-human scenes and performs well.

In summary, the major contributions of our work are:

- We propose Animatable 3D Gaussian, a novel neural representation using 3D Gaussians for dynamic humans, which enables 3D Gaussians to reconstruct human avatars from scratch **without any preprocessing** such as SFM[41].
- We present a novel pipeline for human reconstruction that can acquire higher-fidelity human avatars with **lower memory and time consumption (few seconds)** than state-of-the-art methods [5, 26].
- We propose a time-dependent ambient occlusion module to reconstruct the dynamic shadows, which allows our method to obtain high-fidelity human avatars from **multi-human scenes** with complex motions and dynamic shadows.

## 2 RELATED WORK

**3D Human Reconstruction.** Reconstructing 3D human has been a popular research topic in recent years. Traditional methods achieve high-fidelity reconstruction by means of depth sensors [9, 14, 19, 42] and dense camera arrays [11, 12], but expensive hardware requirements limit the application of these methods. Recent methods [1, 2, 18, 20, 21, 47, 48] utilized parametric mesh templates as a prior, such as SMPL [33]. By optimizing mesh templates, these methods are able to reconstruct 3D human bodies of different shapes. However, they have limitations in reconstructing details such as hair and fabric.

The emergence of neural radiance fields [35] has made neural representations popular in the field of human reconstruction. Many works [3, 5–8, 10, 13, 15, 22–27, 29, 32, 34, 37, 39, 40] have used neural representations to model 3D human shapes and appearances in a canonical space and then used deformation fields to deform the model into a posed space for rendering. These approaches achieve high-quality reconstruction results but require high memory and time consumption, as they typically require complex implicit representations, deformation algorithms, and volume rendering. One of these works InstantAvatar [26] is fast but suffers from artifacts in synthetic images. Our approach solves the problems of previous methods by combining parametric templates and implicit expressions to achieve fast and high-quality human reconstruction.

**Accelerating Neural Rendering.** Since the rendering speed of vanilla NeRF [35] is very slow, it takes several seconds to obtain an image and more time to train. Recent works [4, 16, 17, 28, 30, 31, 36, 38, 43–45, 49] have been devoted to improving the speed of neural rendering. Plenoxels [16] proposed the utilization of a sparse volume, accompanied by density and spherical harmonic coefficients, for rendering purposes. TensoRF [4] decomposed the voxel grid into an aggregate of vector coefficients, aiming to reduce the size of the model for efficiency. Instant-NGP [36] introduced multi-resolution hash encoding to accomplish fast rendering. 3D-GS [28] represented each scene as a collection of scalable semi-transparent ellipsoids to achieve real-time rendering. These methods are mainly used for the reconstruction of static scenes. For dynamic scenes, FPO[45] presented a novel combination of NeRF, PlenOctree[49] representation, volumetric fusion, and Fourier transform to tackle efficient neural modeling and real-time rendering of dynamic scenes. InstantAvatar [26] applied hash encoding on 3d avatar reconstruction tasks for fast rendering. Such methods require high memory costs and suffer from artifacts. In this paper, we extend 3D-GS from static scenes to dynamic human scenes.

## 3 PRELIMINARY

In this section, we briefly review the representation and pipeline of 3D Gaussian rasterization [28] in Sec. 3.1, and pose-based deformation in Sec. 3.2.

### 3.1 3D Gaussian Rasterization

Kerbl *et al.* [28] proposed to represent each scene as a collection of 3D Gaussians. Each 3D Gaussian $P$ is defined as:

$$P = \{x_0, R, S, \alpha_0, SH\}, \tag{1}$$

where $x_0$ represents the geometric center of a 3D Gaussian distribution, $R$ is a rotation matrix, $S$ is a scaling matrix that scales the Gaussian in three dimensions, $\alpha_0$ denotes the opacity of the center, and $SH$ refers to a set of spherical harmonic coefficients used for modeling view-dependent color distribution following standard practice [16].

The opacity of position $x$ in the vicinity of a 3D Gaussian $P$ is defined as:

$$\alpha(x) = \alpha_0 e^{-\frac{1}{2}(x-x_0)^T \Sigma^{-1}(x-x_0)}, \tag{2}$$

where the covariance matrix $\Sigma$ is decomposed into the rotation matrix $R$ and the scaling matrix $S$:

$$\Sigma = RSS^T R^T. \tag{3}$$

Following the approach of Zwicker *et al.* [52], 3D Gaussians are projected to 2D image space as follows:

$$\Sigma' = PW\Sigma W^T P^T, \tag{4}$$

where $W$ is a viewing transformation and $P$ is the Jacobian of the affine approximation of the projective transformation.

Subsequently, the projected Gaussians are sorted based on their depths and rasterized to neighboring pixels according to Eq. (2). Each pixel receives a depth-sorted list of colors $c_i$ and opacities $\alpha_i$. The final pixel color $C$ is computed by blending N ordered points overlapping the pixel:

$$C = \sum_{i \in N} c_i \alpha_i \prod_{j=1}^{i-1} (1 - \alpha_j). \tag{5}$$

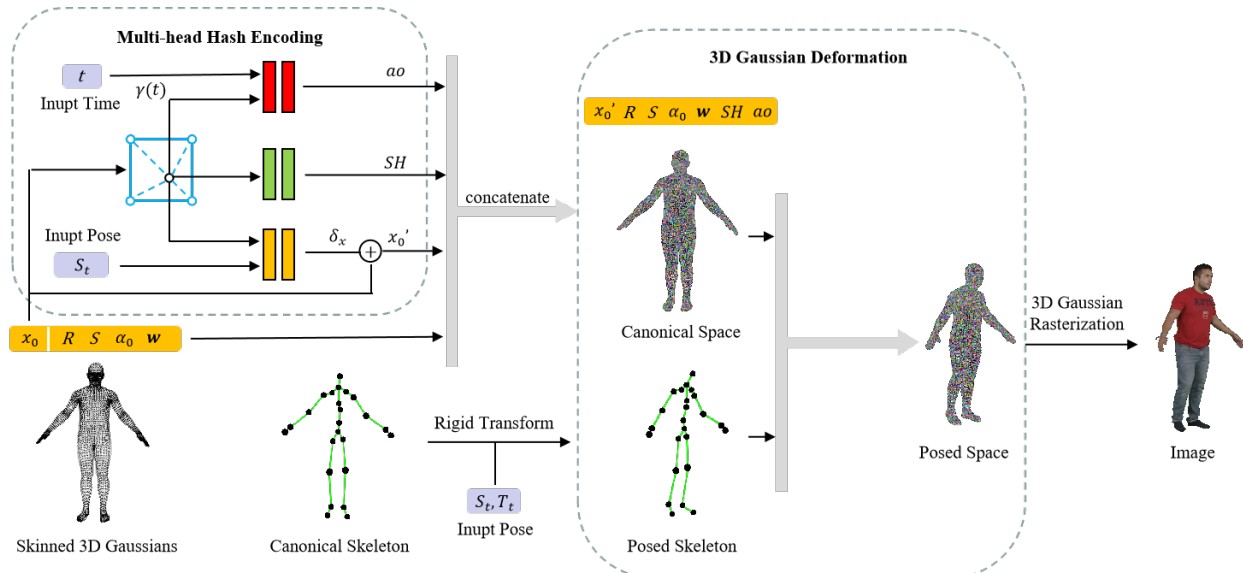

**Figure 2: Overview.** The proposed animatable 3D Gaussian consists of a set of skinned 3D Gaussians and a corresponding canonical skeleton. Each skinned 3D Gaussian contains center $x_0$, rotation $R$, scale $S$, opacity $\alpha_0$, and skinning weights w. First, we sample spherical harmonic coefficients $SH$, vertex displacement $\delta_x$, and ambient occlusion $ao$ from the multi-head hash-encoded parameter field according to the center $x_0$, where the multilayer perceptron for $ao$ requires an additional frequency encoded time $\gamma(t)$ as input. Next, we concatenate the sampled parameters, the original parameters, and a shifted center $x_0^{'}$ in canonical space. Finally, we deform 3D Gaussians to the posed space according to the input pose $S_t, T_t$ and render them to the image using 3D Gaussian rasterization [28].

## 3.2 Pose-guided Deformation

We define each frame within the video sequence as a posed space with a timestamp $t$ and deform points from canonical space to posed space via linear blend skinning. We use a skinning weight field to model articulation. The skinning weight field [33] is defined as:

$$\mathbf{w}(x_c) = \{w_1, ... w_{n_b}\}, \qquad (6)$$

where $n_b$ is the count of bones and $x_c$ is a point in canonical space.

Target bone transformations $\mathbf{B}_t = \{B_1^t, ..., B_{n_b}^t\}$ in frame $t$ can be calculated from the input poses and the corresponding skeleton as follows:

$$\mathbf{S}_t, \mathbf{J}, T_t \mapsto \mathbf{B}_t, \qquad (7)$$

where $\mathbf{S}_t = \{\omega_1^t, ..., \omega_{n_b}^t\}$ refers to the rotation Euler Angle of each joint in frame $t$ (world rotation for $\omega_1^t$ and local rotation for the rest), $T_t$ is the world translation in frame $t$, and $\mathbf{J} = \{J_1, ..., J_{n_b}\}$ is the local position of each joint in canonical space. Please refer to the supplementary materials for the detailed calculation process.

Then we can deform points $x_c$ in canonical space to the posed space via linear blend skinning:

$$x_t = \sum_{i=1}^{n_b} w_i B_i^t x_c, \qquad (8)$$

where $x_t$ refers to a point in frame $t$.

## 4 METHOD

Given a video sequence $\{I_t\}_{t=1}^T$ with one or more moving humans and their poses $\mathbf{S}_t, T_t$, we reconstruct an animatable 3D Gaussian representation for each person in seconds. For multi-view video sequences, we reconstruct time-dependent ambient occlusion to achieve high-quality novel view synthesis.

In this section, we first introduce our animatable 3D Gaussian representation in Sec. 4.1. Then we deform 3D Gaussians from canonical space to posed space, as introduced in Sec. 3.2 and Sec. 4.2. Finally, we run 3D Gaussian rasterization (Sec. 3.1) to render an image for specific camera parameters. Moreover, we propose to build time-dependent ambient occlusion (Sec. 4.3) for multi-view, multi-person, and wide-range motion tasks, which enables our algorithm to fit dynamic shadows caused by occlusion. The pipeline of the proposed method is illustrated in Fig. 2.

### 4.1 Animatable 3D Gaussian in Canonical Space

**3D Gaussians with Skeleton.** We bind 3D Gaussians to a corresponding skeleton to enable the linear skinning algorithm in Eq. (8). The proposed animatable 3D Gaussian representation consists of a set of skinned 3D Gaussians and a corresponding skeleton $\mathbf{J}$ mentioned in Eq. (7). The skinned 3D Gaussian $P_{skin}$ is adapted from the static 3D Gaussian $P$ in Eq. (1) and defined as follows:

$$P_{skin} = \{x_0, R, S, \alpha_0, SH, \mathbf{w}, \delta_x\}, \qquad (9)$$

where $\mathbf{w}$ is the skinning weights, and $\delta_x$ is the vertex displacement.

Directly optimizing the skinning weights from random initialization is a significant challenge. Instead, we treat the skinning weights as a strong prior for a generic human body model unrelated to any specific human shape. In practice, we initialize the skinning weights and positions of Gaussian points using a standard

skinned model as discussed in Sec. 4.4. During optimization, the skinning weights $\mathbf{w}$ are kept fixed, while the vertex displacement $\delta_x$ and skeleton $\mathbf{J}$ are optimized to capture the shape and motion of an individual accurately. We shift the Gaussian center $x_0$ before implementing pose-based deformation in Eq. (8):

$$x_0' = x_0 + \delta_x. \tag{10}$$

**Multi-Head Hash-encoder.** We note that using per-vertex colors performs poorly in deformable dynamic scenes [28]. Since the deformation of Gaussians from canonical space to posed space is initially uncertain, it needs more samples and iterations to reach convergence. Moreover, rendering based on 3D Gaussian rasterization can only backpropagate the gradient to a finite number of Gaussians in a single iteration, which leads to a slow or even divergent optimization process for dynamic scenes. To address this issue, we suggest sampling spherical harmonic coefficients $SH$ for each vertex from a continuous parameter field, which is able to affect all neighboring Gaussians in a single optimization. Similarly, using unconstrained per-vertex displacement can easily cause the optimization process to diverge in dynamic scenes. Therefore, we also model a parameter field for vertex displacement. For the remaining parameters in Eq. (9), we store them in each point to preserve the ability of the 3D Gaussian to fit different shapes. We define the parameter fields as follows:

$$x_0 \mapsto SH. \tag{11}$$

In order to acquire pose-dependent deformation, we add pose $S_t$ as an additional input for vertex displacement:

$$x_0, S_t \mapsto \delta_x. \tag{12}$$

Since our animatable 3D Gaussian representation is initialized by a standard human body model, the centers of 3D Gaussians are uniformly distributed near the human surface. We only need to sample at fixed positions near the surface of the human body in the parameter fields. This allows for significant compression of the hash table for the hash encoding [36]. Thus, we choose the hash encoding to model our parameter field to reduce the time and storage consumption.

Optionally, we provide UV-encoded spherical harmonic coefficients, allowing fast processing of custom human models with UV coordinate mappings. UV encoding potentially achieves higher reconstruction quality compared to hash encoding. The UV mapping for spherical harmonic coefficients $SH$ is defined as follows:

$$u, v \mapsto SH, \tag{13}$$

where $u, v$ is the UV coordinate of a 3D Gaussian.

## 4.2 3D Gaussian Deformation

We use poses to guide the deformation of 3D Gaussians illustrated in Fig. 3. We introduce pose-based deformation which transforms the point position from the canonical space to the posed space in Sec. 3.2. For 3D Gaussians, we also need to deform their orientation to achieve a consistent anisotropic Gaussian distribution at different poses. Using the same orientation in different poses causes the 3D Gaussian to degenerate into Gaussian spheres. Same as Eq. (8), we apply the linear blend skinning to the rotation $R$ of 3D Gaussians

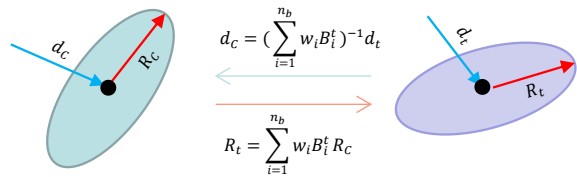

Canonical Space            Posed Space

**Figure 3: 3D Gaussian Deformation. The rotation $R_c$ of 3D Gaussian in canonical space is deformed into the posed space using Eq. (14), while view direction $d_t$ is implemented the inverse transformation in Eq. (15).**

as follows:

$$R_t = \sum_{i=1}^{n_b} w_i B_i^t R_c, \tag{14}$$

where $R_c$ is the rotation in canonical space and $R_t$ is the rotation in posed space in frame $t$.

The view direction for the computation of color based on spherical harmonics should be deformed to canonical space, in order to achieve consistent anisotropic colors. Hence, we apply the inverse transformation of the linear blend skinning to the view direction:

$$d_c = \left(\sum_{i=1}^{n_b} w_i B_i^t\right)^{-1} d_t, \tag{15}$$

where $d_c$ is the direction in canonical space and $d_t$ is the direction in posed space in frame $t$.

We implement the mentioned transformations by extending the 3D Gaussian rasterizer [28] and explicitly derive the gradients for all parameters. This makes the time overhead for the pose-based deformation of 3D Gaussians almost negligible.

## 4.3 Time-dependent Ambient Occlusion

We propose a time-dependent ambient occlusion module to address the issue of dynamic shadows in specific scenes. We additionally model an ambient occlusion factor $ao \in [0, 1]$ on top of the skinned 3D Gaussian in Eq. (9) as follows:

$$P_{skin}' = \{x_0, R, S, \alpha_0, SH, \mathbf{w}, \delta_x, ao\}. \tag{16}$$

We calculate the color after considering the ambient occlusion for each individual Gaussian as follows:

$$c = ao \cdot \mathbf{Y}(SH, d_c), \tag{17}$$

where $\mathbf{Y}$ refers to the spherical harmonics.

As discussed in Sec. 4.1, we also employ hash encoding for the ambient occlusion $ao$, since shadows should be continuously modeled in space. Furthermore, we introduce an additional input of the positional encoding [35] of time $t$ in the MLP module of the hashing encoding [36] in order to capture time-dependent ambient occlusion. The parameter field for ambient occlusion is defined as:

$$x_0, \gamma(t) \mapsto ao, \tag{18}$$

where function $\gamma(\cdot)$ refers to the positional encoding used in NeRF [35].

At the beginning of training, we use a fixed ambient occlusion to allow the model to learn time-independent spherical harmonic

coefficients. We start optimizing the ambient occlusion after the color stabilizes.

## 4.4 Optimization Details

**Animatable 3D Gaussian Initialization.** The initialization of the 3D Gaussian has a significant impact on the quality of optimization results. Improper initialization can even lead to divergence in the optimization process. For static 3D Gaussians [28], the Structure-from-Motion (SFM [41]) method is used to obtain the initial 3D Gaussian point cloud, which provides a very good and dense initialization.

However, for dynamic scenes, obtaining an initial point cloud using SFM is a huge challenge. Therefore, we use a standard skinned model to initialize our deformable 3D Gaussian representation. This standard skinned model should include a set of positions and skinning weights **w**, and a skeleton **J** corresponding to the input poses. We recommend upsampling the vertices of the input model to around 100,000 to achieve high reconstruction quality. Specifically, for the SMPL model [33], we randomly sample K additional points (we set K=20 based on experimental results) within the neighborhood of each vertex and directly copy their skinning parameters to get a model of around 140,000 points.

**Loss Function.** The proposed animatable 3D Gaussian representation is capable of accurately fitting dynamic scenes containing moving humans, thus eliminating the need for additional regularization losses. We directly employ a combination of $\mathcal{L}_1$ and D-SSIM term:

$$\mathcal{L} = (1 - \lambda)\mathcal{L}_1 + \lambda\mathcal{L}_{D-SSIM}, \tag{19}$$

where we use $\lambda = 0.2$ following the best practices of the static 3D Gaussian [28].

## 5 EXPERIMENTS

We evaluate the speed and quality (PSNR, SSIM [46], and LPIPS [51]) of our method on monocular scenes (Sec. 5.1) and multi-view scenes (Sec. 5.2). Comparative experiments with state-of-the-art methods [5, 26] show the superiority of our method. In addition, we provide extensive ablation studies of our methods. In all experiments, we evaluate our approach on a single RTX 3090 and do not use any predicted body parameters but a generic template.

## 5.1 Monocular Scenes

We use the PeopleSnapshot [1] and ZJU-MoCap [37] dataset to evaluate the performance of our method in single-human, monocular scenes.

**PeopleSnapshot.** Following the previous approach [26], we downsample the images to 540×540 resolution and use the SMPL [33] model for initialization. Since this dataset does not contain timestamps and drastic shading variations, we do not use the time-dependent ambient occlusion module.

We provide quantitative comparisons in Tab. 1 and visual comparisons in Fig. 4. Compared to InstantAvatar [26] and Anim-NeRF [5], our method achieves higher reconstruction quality in a shorter training time (5s and 30s). As shown in Fig. 4, our method solves the problem of artifacts that have occurred in the previous methods.

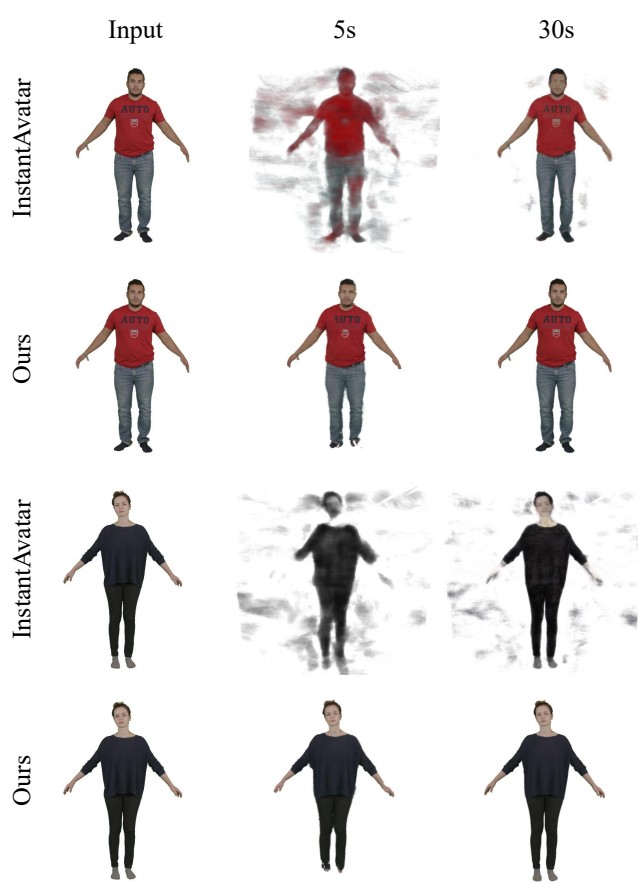

Input        5s        30s

**Figure 4: Qualitative Results on PeopleSnapshot [1] Dataset. We show the image quality of our method and InstantAvatar [26] at 5s and 30s training time. Compared to InstantAvatar, our method achieves higher reconstruction quality and a significant reduction in artifacts.**

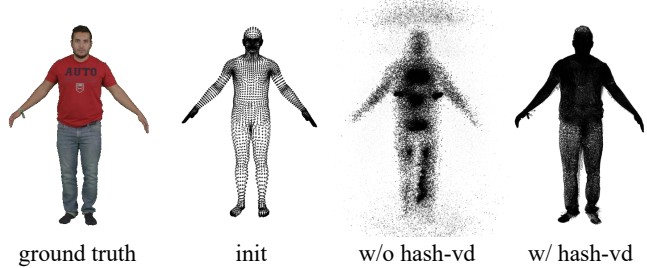

ground truth        init        w/o hash-vd        w/ hash-vd

**Figure 5: Ablation Study of Hash-Encoded Vertex Displacement. Without hash-encoded vertex displacement (w/o hash-vd), the centers of 3D Gaussians may diverge during the optimization process, while our hash-encoded vertex displacement (w/ hash-vd) converges to the ground truth shape.**

**Table 1: Comparison on PeopleSnapshot [1] Dataset. We compare our results with InstantAvatar [26] at 5s and 30s training time and provide Anim-NeRF's results [5] at 5-minute training time.**

|  | training↓ | male-3-casual | | | male-4-casual | | | female-3-casual | | | female-4-casual | | |
|---|---|---|---|---|---|---|---|---|---|---|---|---|---|
|  |  | PSNR↑ | SSIM↑ | LPIPS↓ | PSNR↑ | SSIM↑ | LPIPS↓ | PSNR↑ | SSIM↑ | LPIPS↓ | PSNR↑ | SSIM↑ | LPIPS↓ |
| Anim-NeRF [5] | 5m | 23.17 | 0.9266 | 0.0784 | 22.30 | 0.9235 | 0.0911 | 22.37 | 0.9311 | 0.0784 | 23.18 | 0.9292 | 0.0687 |
| InstantAvatar [26] | 30s | 26.56 | 0.9301 | 0.1190 | 26.10 | 0.9289 | 0.1397 | 22.37 | 0.8427 | 0.2687 | 26.32 | 0.9281 | 0.1333 |
| Ours | 30s | **29.06** | **0.9704** | **0.0264** | **26.16** | **0.9554** | **0.0491** | **24.59** | **0.9535** | **0.0399** | **27.26** | **0.9634** | **0.0281** |
| InstantAvatar [26] | 5s | 17.30 | 0.7980 | 0.3473 | 16.44 | 0.7960 | 0.3508 | 17.22 | 0.8185 | 0.3262 | 14.91 | 0.7606 | 0.3878 |
| Ours | 5s | **22.84** | **0.9395** | **0.0632** | **18.88** | **0.9103** | **0.1175** | **20.51** | **0.9301** | **0.0764** | **20.63** | **0.9246** | **0.0737** |

**Table 2: Comparison on ZJU-MoCap [37]. Compared with competitive baselines [26, 37, 50], our approach achieves the best on SSIM and the best or second best on two other metrics. Cell color indicates best and second best .**

|  | training↓ | FPS↑ | 377 | | | 386 | | | 387 | | | 392 | | | 394 | | |
|---|---|---|---|---|---|---|---|---|---|---|---|---|---|---|---|---|---|
|  |  |  | PSNR↑ | SSIM↑ | LPIPS↓ | PSNR↑ | SSIM↑ | LPIPS↓ | PSNR↑ | SSIM↑ | LPIPS↓ | PSNR↑ | SSIM↑ | LPIPS↓ | PSNR↑ | SSIM↑ | LPIPS↓ |
| NeuralBody [37] | 12h | 2 | 29.11 | 0.9674 | 0.0410 | 30.54 | 0.9678 | 0.0464 | 27.00 | 0.9518 | 0.0594 | 30.10 | 0.9642 | 0.0533 | 29.10 | 0.9593 | 0.0546 |
| MonoHuman [50] | 4d | 0.1 | 29.12 | 0.9727 | 0.0265 | 32.94 | 0.9695 | 0.0361 | 27.93 | 0.9601 | 0.0417 | 29.50 | 0.9635 | 0.0395 | 29.15 | 0.9595 | 0.0381 |
| InstantAvatar [26] | 120s | 8 | 29.92 | 0.9676 | 0.0600 | 32.57 | 0.9605 | 0.0864 | 28.41 | 0.9541 | 0.0861 | 30.98 | 0.9677 | 0.0595 | 30.32 | 0.9617 | 0.0656 |
| Ours | 60s | 120 | 30.51 | 0.9761 | 0.0332 | 32.65 | 0.9733 | 0.0396 | 28.04 | 0.9618 | 0.0526 | 30.77 | 0.9699 | 0.0473 | 29.89 | 0.9626 | 0.0493 |

**Table 3: Ablation Study on PeopleSnapshot [1] Dataset. We respectively remove the hash-encoded spherical harmonic coefficients (hash-SH), hash-encoded vertex displacement (hash-vd), and joint optimization (J-optimize) from our method, and compare their results with our full method.**

|  | PSNR↑ | SSIM↑ | LPIPS↓ |
|---|---|---|---|
| w/o hash-SH | 25.38 | 0.9478 | 0.0504 |
| w/o hash-vd | 22.08 | 0.8994 | 0.1678 |
| w/o J-optimize | 21.45 | 0.9311 | 0.0781 |
| ours | **26.77** | **0.9607** | **0.0359** |

Moreover, our method achieves the fastest training and rendering speed. For 540 × 540 resolution images, we reach a training speed of 50 FPS and a rendering speed of 170 FPS.

**ZJU-MoCap.** We pick 5 sequences (377, 386, 387, 392, 394) in the ZJU-MoCap dataset to evaluate our method and select camera 1 as the train set and the remaining 22 cameras as the test set. In keeping with baselines [26, 37, 50], we downsample the images to 512 × 512 resolution.

Tab. 2 shows that our method is superior to other baseline methods on SSIM, and also achieves best or second best on PSNR and LPIPS. In comparison with NeuralBody [37] and MonoHuman [50], our approach achieves a 240× faster speed in training time and a 60× faster speed in rendering speed. In comparison to the main baseline InstantAvatar [26], our approach achieved a 2× faster speed in training time and a 15× faster speed in rendering speed.

**Ablation Study.** We performed ablation experiments on the PeopleSnapshot dataset. Tab. 3 quantitatively illustrates that our proposed hash-encoded spherical harmonic coefficients, hash-encoded vertex displacement, and joint optimization, can improve the quality of the reconstruction in a short period of training time (30s). We

also visualize the point cloud optimization results in Fig. 5 to show the effect of hash-encoded vertex displacement.

## 5.2 Multi-View Scenes

There are few pose and shadow changes in the PeopleSnapshot [1] dataset. In order to demonstrate the reconstruction performance of the proposed method in scenes containing complex motions and dynamic shadows, we create a dataset called GalaBasketball, which is synthesized from several player models with different shapes and appearances. The GalaBasketball dataset consists of four single-human and three multi-human scenes, providing six uniformly surrounded cameras as a training set and one camera as a test set. For the single-human scenes, we provide an additional set of actions to evaluate the novel pose synthesis capability. Moreover, We provide a standard skinned model corresponding to the GalaBasketball dataset for initialization, which does not resemble any of the players in the dataset in terms of geometry and appearance.

**Novel View Synthesis.** As shown in Tab. 4 and Fig. 6, we evaluate the ability of our method and InstantAvatar [26] to synthesize novel views on the single-human scenes of GalaBasketball. Compared with InstantAvatar [26], our approach under any setting achieves a higher-quality synthesis of novel views in a shorter training time and significantly eliminates artifacts. This proves that our animatable 3D Gaussian representation can be trained under complex motion variations and obtain high-quality human avatars. In contrast, InstantAvatar suffers from artifacts and achieves low synthesis quality, because skin weights fail to converge to the ground truth under complex motion variations.

**Ablation Study.** We also provide the result of raw 3D-GS [28] and our result without ambient occlusion in Tab. 4. The comparison results between 3D-GS and our full methods demonstrate the validity of our proposed Animatable 3D Gaussian representation. The ablation experiment of ambient occlusion shows that the time-dependent ambient occlusion helps our method achieve higher

**Table 4: Comparison on Single-Human Scenes of GalaBasketball Dataset.** We provide our novel view synthesis results under different settings, including without time-dependent ambient occlusion (w/o ao) and our full method, all trained for the 10 epochs (around 1 minute). We compare them with InstantAvatar [26] and raw 3D-GS [28].

| | training↓ | idle | | | dribble | | | shot | | | turn | | |
| --- | --- | --- | --- | --- | --- | --- | --- | --- | --- | --- | --- | --- | --- |
| | | PSNR↑ | SSIM↑ | LPIPS↓ | PSNR↑ | SSIM↑ | LPIPS↓ | PSNR↑ | SSIM↑ | LPIPS↓ | PSNR↑ | SSIM↑ | LPIPS↓ |
| InstantAvatar [26] | 5m | 29.86 | 0.9607 | 0.0575 | 27.25 | 0.9435 | 0.0903 | 29.22 | 0.9461 | 0.0709 | 32.20 | 0.9705 | 0.0371 |
| 3D-GS [28] | 60s | 37.38 | 0.9941 | 0.0042 | 36.53 | 0.9909 | 0.0059 | 37.07 | 0.9908 | 0.0067 | 35.74 | 0.9919 | 0.0071 |
| Ours: w/o ao | 50s | 37.45 | 0.9943 | 0.0046 | 36.87 | 0.9915 | 0.0059 | 37.47 | 0.9921 | 0.0068 | 36.77 | 0.9927 | 0.0062 |
| Ours | 60s | **40.75** | **0.9964** | **0.0029** | **38.53** | **0.9935** | **0.0042** | **39.44** | **0.9936** | **0.0049** | **39.74** | **0.9953** | **0.0038** |

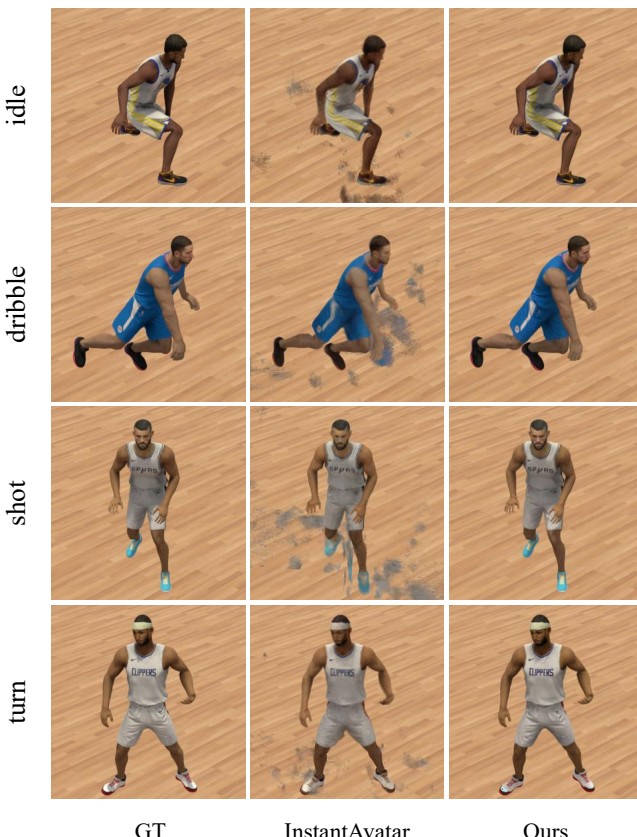

GT     InstantAvatar     Ours

**Figure 6: Novel View Synthesis Results on Single-Human Scenes of GalaBasketball Dataset.** We show the novel view synthesis quality of our method (with hash-encoded spherical harmonic coefficients) and InstantAvatar [26].

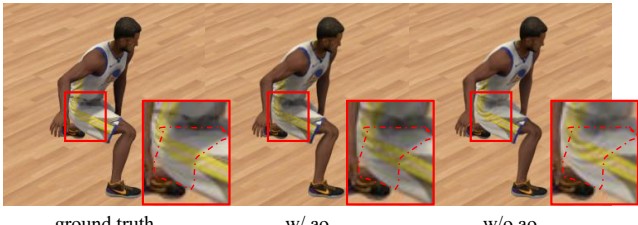

ground truth     w/ ao     w/o ao

**Figure 7: Ablation Study of Time-dependent Ambient Occlusion.** Dynamic shadows can not be reconstructed without time-dependent ambient occlusion (w/o ao), resulting in low novel view synthesis quality. Our proposed time-dependent ambient occlusion is able to fit dynamic shadows and synthesize high-quality novel view shadows.

Novel Pose 1     Novel Pose 2     Novel Pose 3

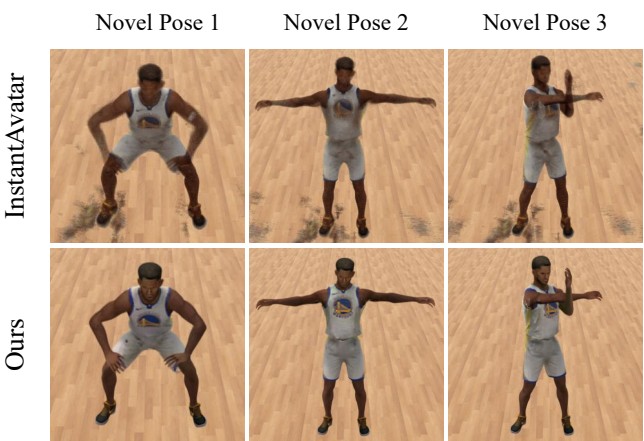

**Figure 8: Novel Pose Synthesis Results on Single-Human Scenes of GalaBasketball Dataset.** We compare the novel pose synthesis quality of our method (without time-dependent ambient occlusion) with InstantAvatar [26].

novel view synthesis quality, which fits time-dependent shadow changes as shown in Fig. 7.

**Novel Pose Synthesis.** We render the images using the other set of actions different from the training set and compare our novel pose synthesis results with InstantAvatar in Fig. 8. Since the novel pose synthesis task does not have a timestamp, we do not use the time-dependent ambient occlusion. The result shows that our

reconstructed human avatar achieves high synthesis quality in novel poses, while InstantAvatar still suffers from artifacts.

**Multi-Human Scenes.** We extend our method to multi-human scenes by rendering multiple human avatars simultaneously on a single RTX 3090, while the existing methods [5, 26] can not be

Novel Pose 1

Novel Pose 2

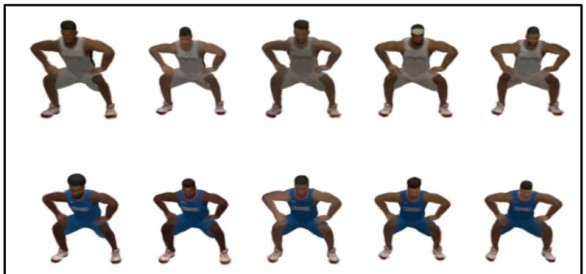

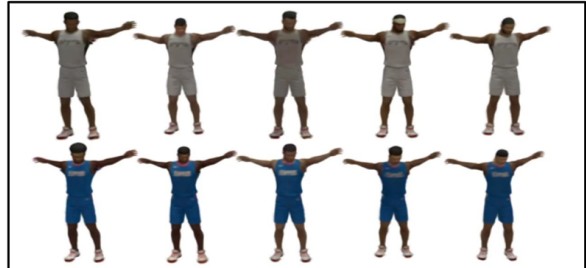

**Figure 9: Novel Pose Synthesis Results on Multi-Human Scenes. We use the 10 human avatars reconstructed from the 5v5 scene to synthesize two sets of novel pose images.**

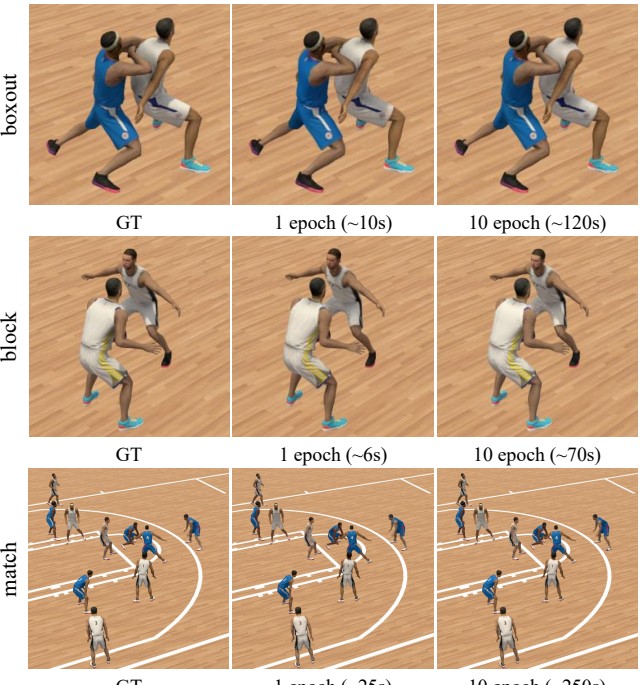

**Figure 10: Novel View Synthesis Results on Multi-Human Scenes. We provide novel view synthesis results of our method trained for 1 epoch and 10 epochs on multi-human scenes. Trained for only one epoch, our method achieves comparable quality to the ground truth (GT). Longer training allows our model to fit higher-quality dynamic shadows.**

extended to multi-human scenes due to the limitations of implementation and memory. Fig. 10 shows the high-quality novel view synthesis results of our method in the multi-human scenes of GalaBasketball, including two double-human scenes and a 5v5 scene. The comparison results between 1 epoch and 10 epochs show that our method can learn the time-dependent dynamic shadows through training. Our method achieves a training speed of 40 FPS and a rendering speed of 110 FPS on double-human scenes ($512 \times 512$

resolution). On the 5v5 scene ($1920 \times 1080$ resolution), our method reaches a training speed of 10 FPS and a rendering speed of 40 FPS.

Our approach reconstructs avatars for each person in the 5v5 scene. Each avatar can synthesize new poses individually. Fig. 9 illustrates the novel pose synthesis results using the ten human avatars reconstructed from the 5v5 scene. Please refer to the supplementary materials for video results.

## 6 CONCLUSION

We propose a method for high-quality human reconstruction in seconds. The reconstructed human avatar can be used for real-time novel view synthesis and novel pose synthesis. Our method consists of the Animatable 3D Gaussian representation in canonical space and pose-based 3D Gaussian deformation, which extends 3D Gaussian [28] to deformable humans. Our proposed time-dependent ambient occlusion achieves high-quality reconstruction results in scenes containing complex motions and dynamic shadows. Compared to the state-of-the-art methods [5, 26], our method takes less training time, renders faster, and yields better reconstruction results. Moreover, our approach can be easily extended to multi-human scenes.

**Limitations and Future Work.** To implement optional modules, our approach uses multiple MLPs. However, in specific tasks, multiple MLPS can be merged into one to further reduce training consumption. Due to the complexity of the dynamic scenes, we do not add or remove Gaussian points during the training process, which leads to the quality of the reconstruction greatly affected by the number of initialized points, and reduces the reconstruction quality in the high-frequency region. Considering the training speed, our method does not optimize pose which fails in scenes with inaccurate poses.

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
