# OpenReview forum: "Animatable 3D Gaussian: Fast and High-Quality Reconstruction of Multiple Human Avatars"
_acmmm.org/ACMMM/2024/Conference — MM2024 Poster_

### Official Review · Reviewer_rLMq · 2024-04-28

**Rating:** 4
**Confidence:** 3

**Summary:**

This paper introduces animatable 3D Gaussians for rapid and high-quality 3D human reconstruction.  The proposed multi-head hash encoding encodes the gaussian  parameters efficiently and the time-dependent ambient occlusion improves the rendering quality a lot on the newly proposed GalaBasketball dataset. The experiments demonstrate that the proposed method requires significantly less training time and GPU memory, leading to faster rendering speeds and improved performance.

**Strengths:**

- The pipeline of the method is straight-forward and effective and the method needs significantly less training time compared with previous work.
- The method not only reconstructs a single human avatar but can also adapt to multi-human situations, which expands its applications.
- The proposed time-dependent ambient occlusion handles dynamic shadows well and synthesizes high-quality novel view shadows.

**Limitations:**

- The paper did not compare its method with other 3D Gaussian-based reconstruction methods. There are other 3D Gaussian methods for human reconstruction that have been accepted by CVPR and have been open-sourced. They also need less training time and memory and achieve fast rendering speed. It seems that these benefits come from 3D Gaussians, not your method.
- The paper only compares its method with InstantAvatar qualitatively. Other nerf-based methods such as instantNVR (Learning Neural Volumetric Representations of Dynamic Humans in Minutes, CVPR2023) should also be compared qualitatively and quantitatively.
- I would like to know how your model performs on videos in the wild, as opposed to those from datasets. How robust is it, and what are the results like for videos with loose clothing?

**Suitability:**

3

---

### Official Review · Reviewer_MfMS · 2024-05-15

**Rating:** 5
**Confidence:** 3

**Summary:**

This paper introduces a novel framework for dynamic and animatable human representation, leveraging skinned 3D Gaussians and pose-dependent deformation strategy on 3D Gaussians. This approach benefits from the efficiency inherent to 3D Gaussian models. To overcome the sparsity of 3D Gaussian optimization and accelerate training, the authors also incorporate a multi-head hash encoder for pose-dependent shape and appearance, as well as a time-dependent ambient occlusion. This ambient occlusion is specifically designed to effectively model scenes with complex motions.

**Strengths:**

I concur with the authors' key idea that 3D Gaussians are an effective representation for dynamic humans due to their learning and rendering efficiency. To utilize 3D Gaussians for human representation, the authors have:

1. Introduced animatable 3D Gaussians, which integrate 3D Gaussians with a skeletal structure to facilitate the use of linear skinning algorithms.
2. Enhanced traditional articulated neural body representations like InstantAvatar by considering vertex displacement in addition to pose-guided deformation through linear blend skinning, which improves the accuracy of shape and motion capture.
3. Developed a time-dependent ambient occlusion that is particularly effective in scenes with complex motion and dynamic shadows.
4. Demonstrated the method's effectiveness and efficiency through experiments on monocular scenes, multiview settings, and innovative multi-human scenes.
5. Conducted a well-structured ablation study that validates the impact of each component within the proposed method.

**Limitations:**

1. In my opinion, the initialization of the 3D Gaussian for human representation without Sfm, which leverages the SMPL mesh body model, is straightforward and should not be considered a unique contribution of this paper.

2. Regarding the experiments on the ZJU Mocap dataset, the authors chose to evaluate five sequences. However, in the original papers for the baseline methods NeuralBody [37] and MonoBody [50], all nine sequences in the dataset were used for evaluation. Therefore, it would be more comprehensive to present results for all sequences in the dataset unless there are specific reasons for the selection.

**Suitability:**

3

---

### Official Review · Reviewer_UoYU · 2024-06-03

**Rating:** 3
**Confidence:** 4

**Summary:**

The paper presents a new method called Animatable 3D Gaussian for reconstructing high-quality drivable human avatars. Traditional neural radiance fields can create good reconstructions but are computationally expensive and not well-suited for scenes with multiple humans and complex shadows. The proposed method overcomes these limitations by learning from input images and poses. The method outperforms a previous methods in terms of reconstruction quality, while also requiring significantly less training time, using less GPU memory and offering faster rendering speed.

**Strengths:**

The authors introduce a new neural representation called Animatable 3D Gaussian, which is more efficient in terms of memory and time compared to existing methods. They also propose a module for time-dependent ambient occlusion that helps recreate dynamic shadows, enhancing the realism of avatars in scenes with multiple moving humans and varying lighting conditions.

**Limitations:**

1. **Inadequate contribution claim** The authors claim they method that  "enables 3D Gaussians to reconstruct human avatars from scratch without any preprocessing such as SFM". However, their method still requires preprocessing steps to acquire per-frame skeleton poses. It's inadequate to claim that no preprocessing is required.
2. **(Possible) Unfair comparisons.**
Table 1 should also show converged results from various methods for fairness. Additionally, the rendering speeds in Table 2 (8 fps) do not match those reported by InstantAvatar(15fps). In figure 6 and 8, the performance of InstantAvatar is quite poor compared to what's reported in their original paper, where it's shown to handle a range of novel poses. If this discrepancy is due to the InstantAvatar not fully converging in the comparison, please ensure to compare after it has converged for accurate results.
3. **Lack of real image comparisons for novel view and novel poses.** The paper lacks a comparison of novel poses and novel views on real data; please provide the relevant results on PeopleSnapshot for fair comparison with previous methods.

**Suitability:**

3

---

### Meta-Review · Area_Chair_4U2C · 2024-06-27

**Recommendation:** Accept (Poster)
**Confidence:** 5

**Metareview:**

All 3 reviewers suggested acceptance for this paper. The AC concurs that this is a quality and timely paper addressing human reconstruction and rendering with animatable 3D Gaussians. Congratulations and pls revise the paper per the reviewer's comments and incorporate the results in the rebuttal file.